# Adaptive Neuromuscular Co-Contraction Strategies Under Varying Approach Speeds and Distances During Single-Leg Jumping: An Exploratory Study

**DOI:** 10.3390/life15121859

**Published:** 2025-12-04

**Authors:** Wei-Hsun Tai, Hsien-Te Peng, Jian-Zhi Lin, Po-Ang Li

**Affiliations:** 1School of General Education, Guangzhou Institute of Science and Technology, Guangzhou 510540, China; dlove520@hotmail.com; 2Graduate Institute of Sport Coaching Science, Chinese Culture University, Taipei 11114, Taiwan; pxd@ulive.pccu.edu.tw; 3Department of Physical Education, Chinese Culture University, Taipei 11114, Taiwan; 4Department of Physical Education, National Taiwan University of Sport, Taichung 404401, Taiwan; 5School of Physical Education, Quanzhou Normal University, Quanzhou 362000, China; 6Department of Civic Education and Leadership, National Taiwan Normal University, Taipei 106308, Taiwan

**Keywords:** single-leg jump, electromyography, co-contraction ratio, neuromuscular control

## Abstract

Purpose: This study investigated how variations in approach speed and distance influence lower-limb muscle activation, joint co-contraction ratios (CCRs), and mechanical joint stiffness during single-leg approach run jump landings (ARJSL), to clarify adaptive neuromuscular strategies for joint stiffness regulation. Methods: Twenty-five physically active male university students performed ARJSLs under six randomized conditions combining two approach speeds (fast > 4.0 m/s; slow < 4.0 m/s) and three approach distances (3, 6, and 9 m). Surface electromyography (sEMG) from five dominant-limb muscles—rectus femoris, biceps femoris, tibialis anterior, gastrocnemius, and soleus—was analyzed across three movement phases: pre-activation, downward (braking), and push-off. Knee and ankle CCRs were computed, while kinematic and kinetic data were used to calculate mechanical joint stiffness via inverse dynamics. A two-way repeated-measures ANOVA evaluated the main and interaction effects of approach speed and distance. Results: Significant speed × distance interactions were observed for tibialis anterior activation, several CCRs, and eccentric ankle stiffness (*p* < 0.05). Pre-activation knee CCR increased with longer, faster approaches, indicating anticipatory joint pre-stiffening. During braking, greater ankle co-contraction under fast–9 m conditions coincided with reduced mechanical ankle stiffness, suggesting a compensatory yielding strategy under high kinetic loads. In the push-off phase, faster approaches elicited higher concentric stiffness at the hip and ankle, supporting efficient energy transfer. Rectus femoris and gastrocnemius activation scaled with both approach speed and distance. Conclusions: Athletes adapt neuromuscular co-contraction and mechanical stiffness in a coordinated, phase-dependent manner to balance protection and performance. These insights may inform targeted training strategies for enhancing jump efficiency and mitigating ACL injury risk.

## 1. Introduction

Single-leg jumping maneuvers are fundamental in sports such as basketball and volleyball, where athletes frequently rely on rapid transitional movements to generate vertical impulse while maintaining postural stability [1,2]. Variations in approach speed and distance are commonly adopted to optimize performance in response to tactical demands [3] such as lay up or spike jump. These external constraints alter the mechanical loading environment of the lower limb, potentially shifting the neuromuscular strategies required to stabilize joints and generate effective propulsion [4].

Emerging evidence indicates that increased approach velocity can enhance vertical jump height by amplifying stretch–shortening cycle utilization and elastic energy storage [5]. However, elevated momentum also imposes greater challenges to joint stability during the braking phase, where athletes must rapidly modulate lower-limb stiffness to prevent excessive knee valgus or anterior tibial shear factors strongly associated with anterior cruciate ligament (ACL) injury risk [6]. Conversely, shorter approach distances limit the development of horizontal velocity, often requiring compensatory increases in muscle activation to maintain comparable performance [7].

Electromyography (EMG) has been widely applied to quantify neuromuscular control strategies during dynamic tasks [8]. In particular, joint co-contraction—the simultaneous activation of antagonist muscle groups surrounding a joint—serves as a protective mechanism to augment joint stiffness, improve proprioceptive acuity, and stabilize kinetic chain alignment under perturbations [9]. Excessive co-contraction, however, may increase metabolic cost and reduce movement efficiency [10]. Despite its relevance, the modulation of joint co-contraction in response to changes in approach speed and distance during single-leg jumping remains insufficiently understood [11].

Previous studies have primarily focused on lower-limb kinematics and kinetics, providing valuable insight into landing mechanics and joint loading patterns [12,13,14,15]. Yet, fewer investigations have examined how neuromuscular activation patterns adapt to varying approach conditions, particularly within a single-leg context where dynamic balance demands are amplified. Understanding these adaptive strategies may offer critical implications for performance enhancement, fatigue-resistant technique development, and ACL injury prevention [16,17].

The primary objective of this study was to clarify the phase specific neuromechanical control strategies athletes employ to regulate joint stability during the ARJSL. Based on the known challenges of momentum management, we tested two hypotheses: (1) The CCR and mechanical stiffness (K) of the knee and ankle joints will increase with greater approach speed and distance, particularly during the eccentric braking phase. (2) The relationship between CCR and K will be influenced by different movement phases such as downward or takeoff.

## 2. Materials and Methods

### 2.1. Participants and Experimental Design

Twenty-five competitive male volleyball players (university Division-II varsity team) participated in this study (age: 21.9 ± 1.5 years; height: 1.80 ± 0.06 m; body mass: 71.9 ± 8.2 kg; volleyball training experience: 7.2 ± 1.8 years). Participants trained four days per week with each session lasting 2–3 h (total weekly training volume 9.5 ± 1.4 h), consisting primarily of technical/tactical volleyball practice, basic strength conditioning, and on-court jump training. All participants reported no history of lower-limb musculoskeletal injury within the preceding six months and had prior exposure to plyometric or jump-based physical training. Limb dominance was defined as the preferred take-off leg during single-leg jumping tasks. The study protocol was approved by the institutional review board of National Taiwan University (protocol code: 2014ES086), and all experimental procedures were conducted in accordance with the Declaration of Helsinki. Written informed consent was obtained from all participants prior to participation.

A within-subject repeated-measures factorial design was employed, incorporating two approach speeds (fast, >4.0 m/s; slow, <4.0 m/s) and three approach distances (3 m, 6 m, and 9 m), resulting in six single-leg approach running jump (ARJSL) conditions. An LED light bar system synchronized to 4 m/s was positioned beside the runway to aid in controlling approach speed and guide participants toward the target ranges. The order of testing conditions was randomized to minimize potential learning or fatigue effects. Participants performed a standardized 10 min dynamic warm-up before testing, designed to minimize fatigue while ensuring neuromuscular readiness. The warm-up included low-intensity jogging on the treadmill, static and dynamic stretching of the lower limb muscles, and submaximal practice jumps. The practice jumps were specifically performed to ensure familiarization with the required approach speed, landing location, and maximal vertical jump effort for the day’s testing protocol. For each condition, a minimum of three successful trials were collected, with successful execution defined by stable foot placement on the force platform and approach velocity within pre-established target ranges.

### 2.2. Instrumentation, Setup, and Synchronization

Three-dimensional kinematic data were acquired using a multi-camera infrared motion capture system (Motion Analysis Corporation, Santa Rosa, CA, USA; 200 Hz). Ground reaction forces were simultaneously recorded using a floor-embedded force platform (AMTI, Watertown, MA, USA; 1000 Hz), positioned flush with the laboratory runway to prevent any disturbance during take-off. Hardware-based synchronization was employed to ensure precise temporal alignment among kinematic, kinetic, and electromyographic signals. The synchronized motion and force data were used to perform inverse dynamics analyses, from which three-dimensional joint moments at the hip, knee, and ankle were calculated [18]. These joint moments, in conjunction with corresponding joint angular displacements, were utilized to compute mechanical joint stiffness (k = ΔM/Δθ) during the landing and take-off sequences [19]. Vertical ground reaction force (vGRF) profiles were used to identify three key temporal instants: initial contact (IC), peak vGRF, and toe-off (TO). These events defined the phases of analysis for electromyographic activity, joint coordination, and mechanical stiffness evaluation. Eccentric Phase was from IC to peak vGRF, and concentric phase was from peak vGRF to TO.

### 2.3. Surface Electromyography (sEMG) Acquisition and Processing

Wireless sEMG signals (Trigno, Delsys, Natick, MA, USA, 2000 Hz) were recorded from five muscles of the dominant limb (Figure 1): rectus femoris (RF), biceps femoris (BF), tibialis anterior (TA), gastrocnemius (GA), and soleus (SOL). Following the SENIAM guidelines, electrode sites were shaved, abraded, and cleansed with 70% alcohol to minimize skin impedance. Bipolar Ag/AgCl electrodes were positioned parallel to the muscle fibers with a 20 mm inter-electrode distance, and a reference electrode was placed over the patella.

Raw sEMG data were processed using a fourth-order Butterworth band-pass filter (4th-order zero-lag Butterworth, 20–450 Hz) to remove movement artifacts and high-frequency noise [20]. Following filtering, all trials were manually inspected for residual artifacts; those containing visible movement artifacts, baseline drift, or non-physiological spikes were excluded. The retained signals were then full-wave rectified and smoothed using a 50 ms root-mean-square (RMS) moving window. Amplitude normalization was performed relative to the peak RMS value obtained during a maximal voluntary countermovement jump with arm swing, expressed as a percentage of reference voluntary contraction (%RVC) [21]. This dynamic reference task was selected because it closely mimics the multi-joint, high-velocity extensor activation pattern observed during the experimental spike-jump landings, elicits higher peak EMG amplitudes in lower-limb musculature than isometric MVIC in athletic populations, and enhances ecological validity when the primary research focus is neuromuscular control during rapid, sport-specific deceleration tasks [22]. Three analysis phases were defined relative to the vGRF profile to characterize muscle activity throughout the landing and take-off sequence (Figure 2). The pre-activation phase was defined as the 100 ms interval preceding IC. The downward phase extended from IC to the point of peak vGRF. Finally, the push-off phase spanned from peak vGRF to TO. Phase boundaries were determined on a trial-by-trial basis; consequently, absolute phase durations varied across participants and experimental conditions. To preserve the natural timing of rapid neuromechanical events (particularly pre-activation and early braking-phase stiffness regulation), no time normalization was applied.

Muscle activation onset timing was identified as the instant when the RMS-processed signal exceeded 150% of the baseline during quiet standing for at least 20 ms. To evaluate joint stabilization strategies, co-contraction ratios (CCR) were computed for both the knee (RF/BF) and ankle (TA/GA) joints.

### 2.4. Statistical Analysis

For each of the six experimental conditions (3 approach distances × 2 instructed speeds), participants performed three valid trials. All participants contributed three successful trials per condition, yielding 75 total trials (n = 25 participants × 3 trials) for each condition. All dependent variables were averaged across the three trials for each participant and condition prior to statistical analysis. The averaged data were then analyzed using two-way repeated-measures ANOVA to examine the main effects of approach distance (3 m, 6 m, 9 m) and speed (slow, fast) and their interaction. Sphericity was assessed with Mauchly’s test; Greenhouse–Geisser correction was applied when violated. Significant main or interaction effects were followed by Bonferroni-adjusted post hoc pairwise comparisons. Effect sizes are reported as partial eta squared (η^2^). The significance level was set at α = 0.05. All statistical analyses were performed using SPSS (Version 28.0; IBM Corp., Armonk, NY, USA). The final sample size for every variable and every experimental condition was therefore n = 25.

## 3. Results

Table 1 presents the mean ± SD values of normalized muscle activation (%RVC) and joint CCR (%) across three distances (3 m, 6 m, and 9 m) under slow and fast speed conditions. The actual measured approach speeds were 3.81 ± 0.31 m/s for the slow condition and 4.25 ± 0.28 m/s for the fast condition. Most muscle activation variables (RF, BF, GA, SOL) showed no significant interaction between speed and distance. However, significant interaction effects were found for the TA activation and several co-contraction ratios (preKCR, downKCR, downACR, pushKCR, pushACR), indicating that changes in speed and distance jointly influenced lower limb muscle coordination patterns.

TA exhibited a significant speed × distance interaction (*F*_(2,48)_ = 9.91, *p* = 0.002, *η*^2^ = 0.25). Among the experimental conditions, the lowest activation occurred during the slow/3 m approach (165 ± 92 %RVC), whereas the highest activation was observed at fast/9 m (423 ± 331 %RVC). Post hoc comparisons revealed that TA activation was significantly greater in the fast than in the slow condition at both 3 m (*p* = 0.001) and 9 m (*p* = 0.002). Furthermore, irrespective of approach speed, 6 m and 9 m approaches elicited significantly higher TA activity than 3 m (Figure 3).

RF demonstrated a main effect of distance (*F*_(2,48)_ = 4.41, *p* = 0.019, *η*^2^ = 0.07), with activation significantly greater at 6 m and 9 m compared to 3 m (Figure 4a). GA activation was greater during the fast approach compared with the slow condition (*F*_(1,24)_ = 9.83, *p* = 0.004, *η*^2^ = 0.29), and increased with approach distance, showing higher activation at 6 m compared to 3 m (*F*_(2,48)_ = 5.34, *p* = 0.012, *η*^2^ = 0.32) (Figure 4b). Neither BF nor SOL exhibited significant main effects or interactions in the phase-averaged activation (% RVC) across the 2 × 3 design (BF: interaction *p* = 0.442, distance *p* = 0.171; SOL: speed *p* = 0.473, distance *p* = 0.056).

Figure 5 shows the muscle co-contraction ratios (%) before ground contact that preKCR showed a significant speed × distance interaction (*F*_(2,48)_ = 9.45, *p* = 0.001, *η*^2^ = 0.28). Simple-effect analyses indicated a crossover pattern: preKCR was significantly lower in the fast condition than in the slow condition at the 3 m distance (*F*_(1,24)_ = 6.03, *p* = 0.021). Conversely, preKCR was significantly higher in the fast condition at both 6 m (*F*_(1,24)_ = 19.57, *p* < 0.001) and 9 m (*F*_(1,24)_ = 8.10, *p* = 0.009).

Post hoc tests revealed that within the slow condition, 3 m resulted in significantly higher preKCR than 6 m (*p* = 0.004). In the fast condition, both 6 m and 9 m distances produced significantly higher preKCR than 3 m (*p* = 0.024 and *p* = 0.029, respectively). No significant interaction was found for preACR (*F*_(2,48)_ = 1.39, *p* = 0.261). Neither the main effect of speed (*F*_(1,24)_ = 1.33, *p* = 0.261, nor distance, *F*_(2,48)_ = 2.45, *p* = 0.108) reached statistical significance.

DownKCR showed a significant speed × distance interaction (*F*_(2,48)_ = 15.52, *p* < 0.001, *η*^2^ = 0.39). Simple-effect analyses showed that downKCR was significantly higher in the fast condition than in the slow condition at 6 m (*p* < 0.001), but lower in the fast condition at 9 m (*p* = 0.009). Within the slow condition, 9 m produced significantly higher downKCR than both 3 m and 6 m (both *p* < 0.001) (Figure 6a). downACR also exhibited a significant speed × distance interaction (*F*_(2,48)_ = 17.72, *p* < 0.001, *η*^2^ = 0.43). Simple-effect analyses revealed a reversed pattern: DownACR was significantly lower in the fast condition than in the slow condition at 3 m (*p* < 0.001) and 6 m (*p* = 0.001), but significantly higher in the fast condition at 9 m (*p* < 0.001) (Figure 6b).

PushKCR demonstrated a significant speed × distance interaction (*F*_(2,48)_ = 3.65, *p* = 0.004, *η*^2^ = 0.29). The only significant simple effect of Speed was observed at 6 m, where the fast condition was significantly higher than the slow condition (*F*_(1,24)_ = 9.97, *p* = 0.004). Within the slow condition, pushKCR was significantly lower at 6 m than at both 3 m and 9 m (both *p* < 0.001). No significant effect of distance was found for the fast condition (*p* = 0.054) (Figure 7a). PushACR also showed a significant speed × distance interaction (*F*_(2,48)_ = 10.2, *p* < 0.001, *η*^2^ = 0.50). Simple-effect analyses indicated that PushACR was significantly higher in the fast condition than in the slow condition at 3 m (*p* < 0.001) and 6 m (*p* = 0.044). No significant difference was observed between speeds at 9 m (*p* = 0.476). The main effect of Distance was not significant for either the slow (*p* = 0.416) or fast (*p* = 0.376) condition (Figure 7b).

In the eccentric phase, hip stiffness showed a significant main effect of distance (*p* = 0.001), with post hoc analysis indicating that the longer approaches resulted in significantly higher hip stiffness compared to the shortest approach (Figure 8). Ankle stiffness was the only joint to exhibit a significant three-way pattern: a significant interaction (*p* = 0.003).

The simple main effects showed that ankle stiffness was significantly greater in the fast condition compared to the slow condition specifically at the 9 m approach and stiffness at the 9 m approach was significantly greater than both the 3 m and 6 m approaches (Figure 9). Knee stiffness showed no significant main effects or interaction during the eccentric phase.

During the concentric phase, both the Hip and Ankle joints showed significant main effects of speed (Figure 10). The concentric stiffness of the Knee joint showed no significant effects.

Table 2 presents the results of the two-way repeated measures ANOVA, and Table 3 provides the mean ± SD values for joint stiffness across the six experimental conditions.

## 4. Discussion

This study examined the effects of approach speed and distance on lower-limb muscle activation and joint CCR during ARJSL. The findings revealed that CCR were consistently and strongly modulated by the interaction between approach speed and distance. Importantly, these neuromuscular adaptations were corroborated by mechanical joint stiffness analyses, demonstrating that variations in CCR were directly reflected in corresponding changes in joint mechanics. Collectively, these results indicate that athletes employ refined, context-dependent stabilization strategies to dynamically regulate joint stiffness and inter-joint coordination, thereby optimizing control and energy management under varying linear momentum demands during high-intensity landings.

The significant interaction found for preKCR indicates a non-linear anticipatory modulation aligned with the predicted kinetic load. This neuromuscular adjustment was accompanied though not always statistically reflected by corresponding variations in mechanical knee stiffness (Table 3). Contrary to the initial expectation that higher approach speeds would uniformly elevate preparatory co-contraction, the results revealed a clear context dependence. At the 3 m distance, preKCR was lower under the fast condition, suggesting reduced preparatory stiffness that may allow greater early-phase compliance to attenuate the initial impact peak [23]. Notably, despite the reduced preKCR, eccentric knee stiffness was numerically higher in the fast than the slow condition at 3 m, indicating that preparatory modulation does not directly predict the magnitude of mechanical stiffness during the eccentric phase. At longer distances (6 m and 9 m), preKCR increased substantially under fast approaches, reflecting enhanced pre-stiffening to accommodate higher momentum and greater eccentric loading demands [24,25]. However, knee stiffness at 9 m–fast did not exceed that at 3 m–fast, demonstrating that the relationship between CCR and mechanical stiffness is not linear, and increases in neural co-contraction do not necessarily yield proportionally higher joint rigidity.

During the downward (braking) phase, clear modulation of co-activation patterns indicated a redistribution of stabilization demands along the lower limb. The marked increase in TA activation and downACR in the fast–9 m condition highlights a shift toward greater distal involvement under high-velocity landing demands [26]. However, eccentric ankle stiffness showed a significant speed × distance interaction (Table 2), diverging from the activation pattern. Although downACR was highest in the fast–9 m condition, mechanical ankle stiffness was lowest in this same condition and significantly highest in the slow–9 m condition (Table 3). This discrepancy indicates that elevated neural co-contraction does not necessarily yield proportionally higher mechanical stiffness, particularly when external loading exceeds the joint’s capacity to resist angular displacement. At shorter distances, lower downACR under fast conditions suggests that high distal stiffening is only required when both approach speed and momentum are elevated [27]. The interaction observed in downKCR further supports a redistribution of braking demands: under slower approaches, the knee contributed more to joint stiffness, whereas faster approaches shifted the control emphasis distally toward the ankle, reflecting coordinated regulation of energy absorption across joints [24].

During the push-off phase, the observed variations in joint CCR mark a functional transition from impact absorption to force generation [28]. This pattern aligned with the concentric stiffness results, as both hip and ankle stiffness demonstrated significant main effects of speed, with consistently greater rigidity under fast approaches (Table 2 and Table 3). These increases indicate that higher approach velocities require a stiffer limb configuration for effective force transmission during propulsion. The elevated pushKCR in the 6 m–fast condition suggests an increased demand for knee stabilization in this intermediate configuration, consistent with the need to convert braking forces into propulsive output [29]. The higher pushACR observed in the fast 3 m and 6 m conditions also indicates increased ankle involvement when propulsion occurs from lower approach momenta [30]. At 9 m, the reduced co-contraction despite maximal approach momentum suggests that propulsion relied less on active stiffening and more on elastic energy return and coordinated segmental extension [30,31]. Overall, these results show that propulsion performance is governed by both the magnitude of concentric stiffness and the context-specific coordination of joint co-contraction.

The main effects observed for the RF and GA, both showing activation increases with distance and speed, confirm that muscle intensity scales with task demand [32]. However, the CCR findings indicate that performance regulation is governed more by intermuscular coordination than by the absolute magnitude of activation [33]. Effective joint stabilization and controlled energy transfer therefore depend on the balanced modulation of agonist–antagonist activity, which adjusts joint stiffness and kinematic control throughout the landing–takeoff sequence [34,35]. Overall, the combined results suggest that CCR serves as a phase-specific neural mechanism that enhances eccentric stability during impact and supports efficient force transmission during propulsion, consistent with the mechanical stiffness adjustments observed across conditions.

The present findings have several practical implications for sports involving approach runs followed by single-leg jump and landing tasks, such as volleyball and basketball. Training programs should incorporate variable approach velocities (particularly ≥4 m/s) and distances (especially 5–7 m) to improve athletes’ capacity to modulate pre-landing muscle co-contraction and braking-phase joint stiffness under differing momentum demands, thereby enhancing deceleration control and ACL-protective knee stability. Given the increased ankle contribution under high-momentum conditions, distal joint stabilization is critical; thus, training should emphasize eccentric plantarflexor strengthening, ankle proprioceptive exercises, and perturbation-based landing drills to optimize load distribution and distal stiffness regulation. The tight coupling between co-contraction ratios and mechanical stiffness during the eccentric–concentric transition supports the inclusion of progressive plyometric protocols such as drop jumps, approach jump–landings, and single-leg rebound tasks to develop rapid stiffness modulation, with a practical workload comprising 3–5 sets of 4–6 controlled repetitions with progressively increasing approach distances and 2–3 min of recovery to preserve neuromuscular performance quality. Finally, the non-linear co-contraction responses indicate that no single “optimal” knee angle or approach velocity exists for all athletes; individualized biomechanical assessments of landing mechanics and approach mechanics are therefore recommended to guide personalized technical adjustments, ultimately improving performance while reducing cumulative knee joint loading in athletes exposed to repetitive high-intensity jump–landing sequences.

This study has several limitations that should be acknowledged to appropriately contextualize the findings. The sample consisted exclusively of physically active young adult males, which restricts the generalizability of the results to broader athletic populations. As this study included only males, and females show higher knee injury risk and different neuromuscular control during jump-landing, these findings may not fully apply to female athletes. Additionally, the controlled laboratory environment cannot fully replicate the complexity of sport-specific movement contexts. Future research should incorporate ecologically valid tasks, including fatigue, perturbations, or external instability, to better approximate competitive conditions. Finally, although surface EMG and mechanically derived joint stiffness provided complementary information, surface EMG cannot isolate deep stabilizing muscles or distinguish active co-contraction from passive viscoelastic stiffness, and joint stiffness calculations represent a combined muscular passive response. [36]. Therefore, while CCR was related to changes in stiffness, the precise active neuromuscular contribution remains inferred rather than directly quantified [37]. Integrating inverse dynamics with EMG-driven musculoskeletal modeling in future work may help clarify the kinetic mechanisms underpinning CCR adaptations.

## 5. Conclusions

The findings of this exploratory study suggest that athletes adjust joint stability through coordinated modulation of muscle co-contraction and mechanical stiffness in response to variations in approach speed and distance during single-leg approach run jump landings. The integration of electromyographic and mechanical analyses provides strong evidence suggesting that anticipatory and phase-specific adaptations are essential for maintaining performance and reducing injury risk. Under higher momentum conditions, greater pre-activation and braking-phase co-contraction appear to contribute to increased joint pre-stiffening and distal control, particularly at the ankle. However, the observed reduction in mechanical stiffness under maximal approach conditions may reflect a compensatory yielding mechanism that allows controlled energy absorption and protects joint integrity. During the propulsion phase, the increase concentric stiffness at the hip and ankle seems to support more efficient force transmission and jump performance. Collectively, these results underscore the role of neuromechanical coordination in the regulation of joint stiffness and may inform training strategies designed to optimize landing mechanics and potentially reduce anterior cruciate ligament injury risk.

## Figures and Tables

**Figure 1 life-15-01859-f001:**
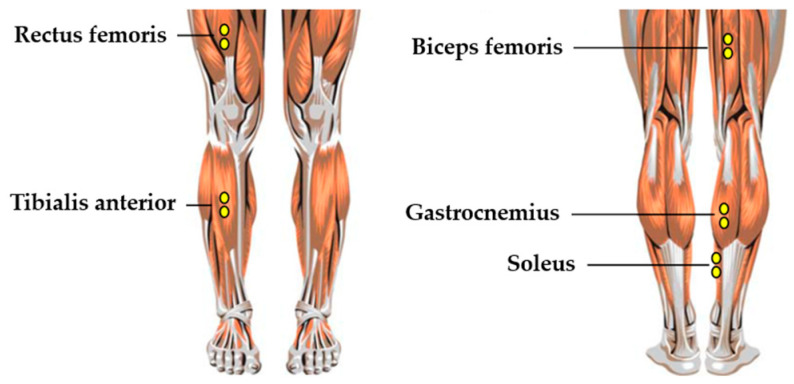
Electrode placement for surface EMG acquisition.

**Figure 2 life-15-01859-f002:**
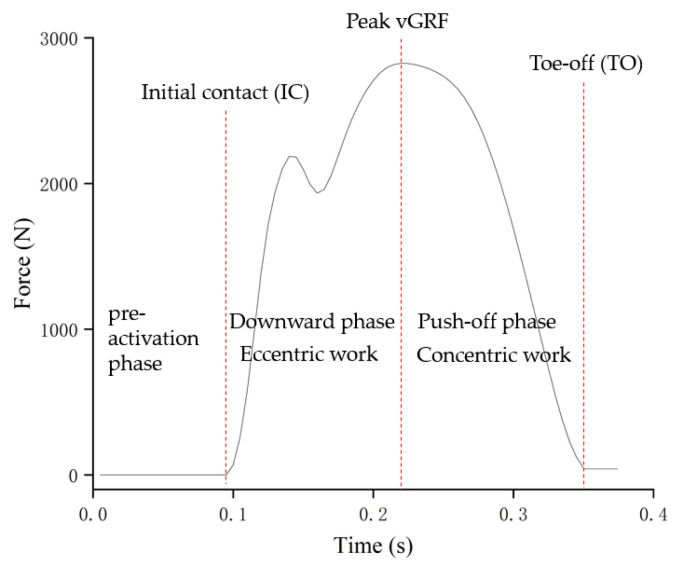
Three analysis phases defined during ARJSL.

**Figure 3 life-15-01859-f003:**
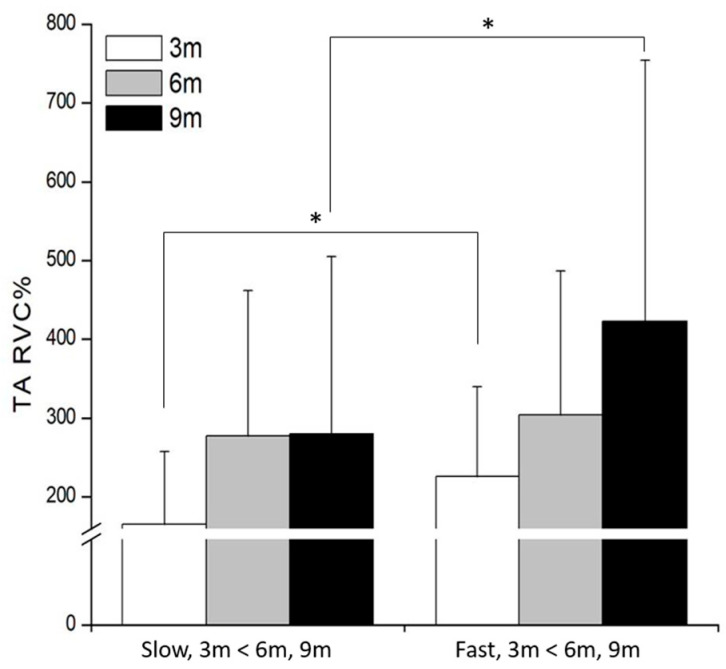
The results of TA muscle activation (%RVC) at different distances and speed conditions. * *p* < 0.05.

**Figure 4 life-15-01859-f004:**
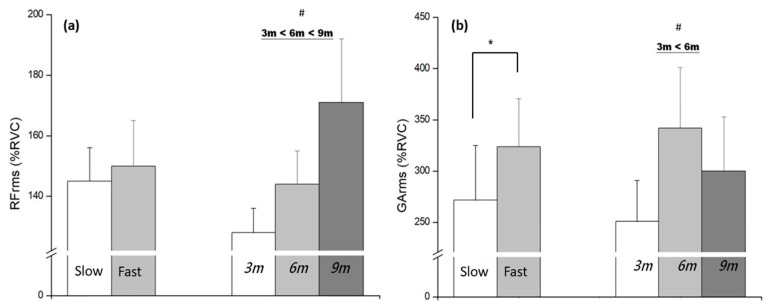
The results of muscle activation (%RVC) at different distances and speed conditions: (**a**) Main effect of RF (**b**) Main effect of GA. * significant difference found between the speeds (*p* < 0.05); # significant difference found between the distance (*p* < 0.05).

**Figure 5 life-15-01859-f005:**
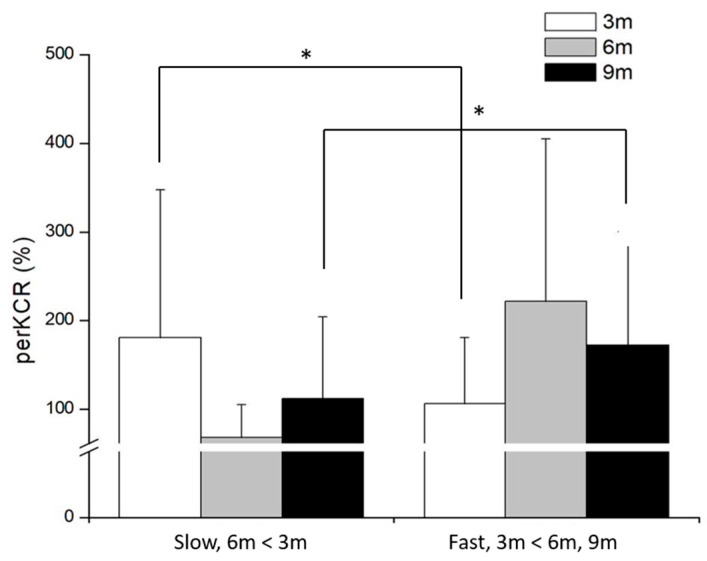
The results of muscle co-contraction ratios (%) before ground contact at different distances and speed conditions. * significant difference found between the speeds (*p* < 0.05).

**Figure 6 life-15-01859-f006:**
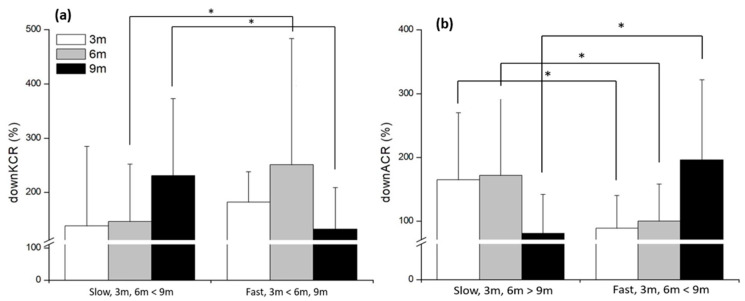
The results of muscle co-contraction ratios (%) at downward phase in different distances and speed conditions. (**a**) Simple main effect of knee (**b**) Simple main effect of ankle. * significant difference found between the speeds (*p* < 0.05).

**Figure 7 life-15-01859-f007:**
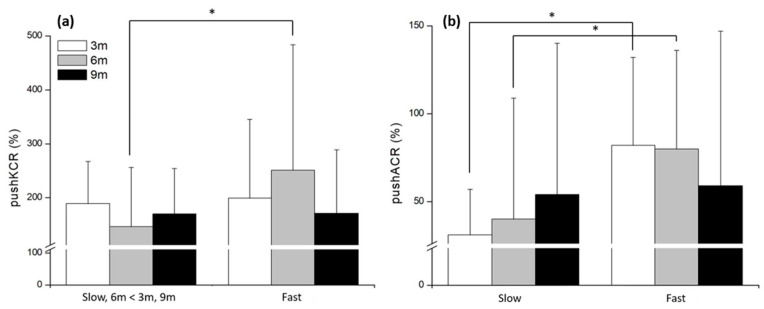
The results of muscle co-contraction ratios (%) at pushoff phase in different distances and speed conditions. (**a**) Simple main effect of knee (**b**) Simple main effect of ankle. * significant difference found between the speeds (*p* < 0.05).

**Figure 8 life-15-01859-f008:**
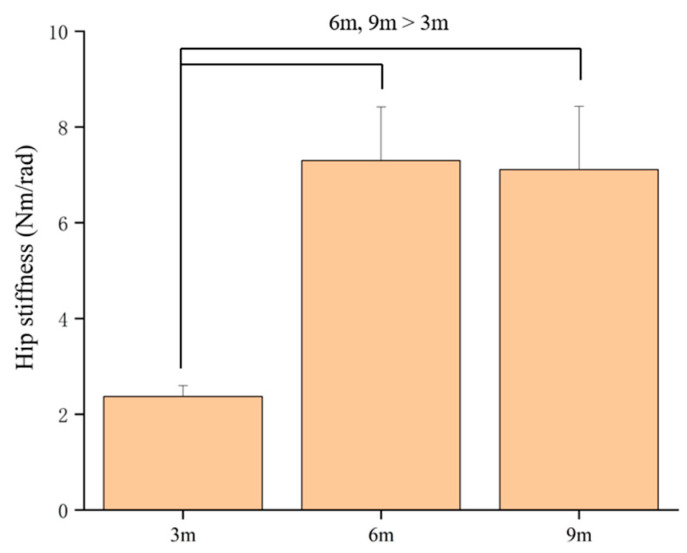
The main effect of distance on eccentric hip stiffness.

**Figure 9 life-15-01859-f009:**
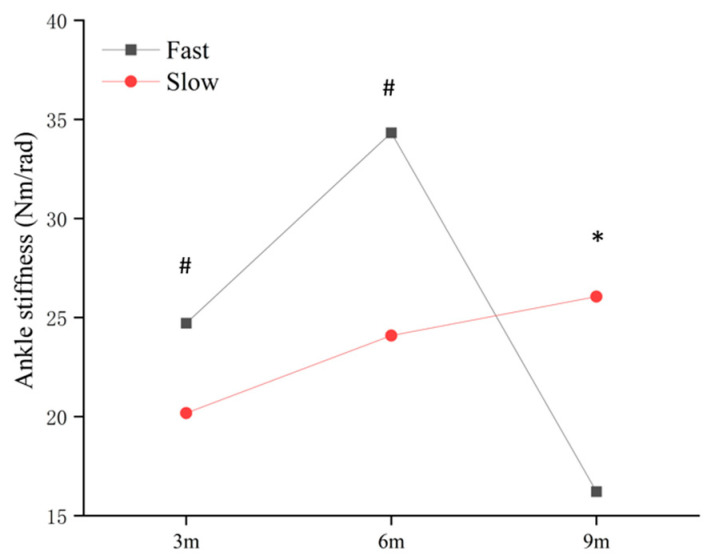
The simple main effect of distance and speeds on eccentric ankle stiffness. * significant difference found between the speeds (*p* < 0.05); ^#^ significant difference found compared with 9 m (*p* < 0.05).

**Figure 10 life-15-01859-f010:**
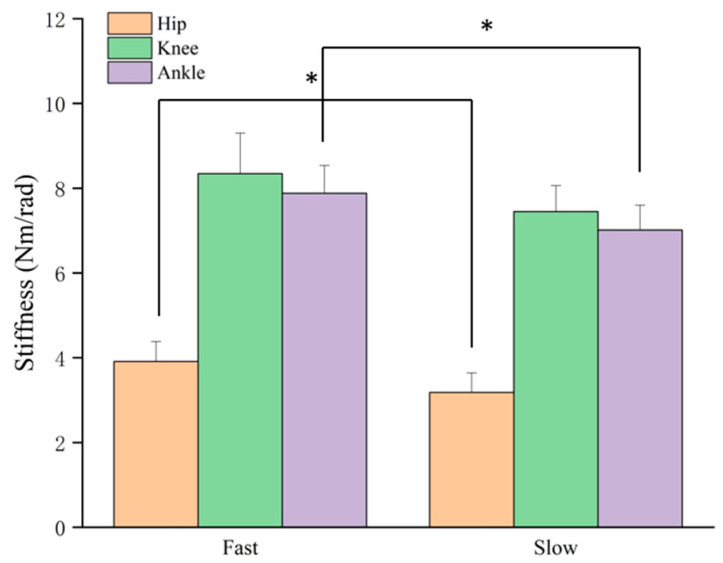
The main effect of lower limb joint stiffness of approach speeds on concentric phase. * *p* < 0.05.

**Table 1 life-15-01859-t001:** Mean ± SD of normalized muscle activation (%RVC) and joint co-contraction ratios (%) at different distances and speeds, and interaction effects.

Variable	Interaction	Speed	3 m	6 m	9 m
RF %RVC	*p* = 0.595	Slow	128 ± 8	144 ± 11	171 ± 21
		Fast	124 ± 10	145 ± 15	144 ± 25
BF %RVC	*p* = 0.442	Slow	145 ± 11	153 ± 10	149 ± 13
		Fast	142 ± 10	136 ± 10	155 ± 16
TA %RVC	*p* = 0.002 *	Slow	165 ± 92	277 ± 185	280 ± 225
		Fast	226 ± 114	304 ± 183	423 ± 331
GA %RVC	*p* = 0.469	Slow	251 ± 40	342 ± 59	290 ± 50
		Fast	272 ± 53	355 ± 62	344 ± 68
SOL %RVC	*p* = 0.476	Slow	145 ± 13	166 ± 12	175 ± 18
		Fast	155 ± 17	177 ± 20	174 ± 19
PreKCR (%)	*p* = 0.001 *	Slow	181 ± 167	68 ± 37	112 ± 92
		Fast	106 ± 75	222 ± 183	172 ± 119
PreACR (%)	*p* = 0.261	Slow	85 ± 88	114 ± 126	88 ± 84
		Fast	96 ± 106	110 ± 103	120 ± 145
DownKCR (%)	*p* < 0.001 *	Slow	138 ± 147	146 ± 106	231 ± 142
		Fast	182 ± 56	251 ± 232	132 ± 77
DownACR (%)	*p* < 0.001 *	Slow	165 ± 105	172 ± 120	81 ± 61
		Fast	89 ± 51	100 ± 58	196 ± 126
PushKCR (%)	*p* = 0.004 *	Slow	189 ± 78	146 ± 110	170 ± 84
		Fast	199 ± 146	251 ± 232	171 ± 118
PushACR (%)	*p* < 0.001 *	Slow	31 ± 26	40 ± 69	54 ± 86
		Fast	82 ± 50	80 ± 56	59 ± 88

* *p* < 0.05.

**Table 2 life-15-01859-t002:** Results of main and interaction effects for lower-limb joint stiffness in eccentric and concentric phases.

Phase	Joint	*p* (Distance)	*p* (Velocity)	*p* (Interaction)	pos hoc
Eccentric	Hip	0.001 *	0.342	0.12	6 m, 9 m > 3 m
	Knee	0.362	0.073	0.289	
	Ankle	0.007 *	0.005 *	0.003 *	9F > 9S; 3 m, 6 m > 9 m
Concentric	Hip	0.056	0.018 *	0.333	F > S
	Knee	0.086	0.159	0.689	
	Ankle	0.105	0.012 *	0.257	F > S

* *p* < 0.05; F: fast approach, S: slow approach.

**Table 3 life-15-01859-t003:** Mean ± SD of joint stiffness at different distances and speeds.

	3 m		6 m		9 m	
Joints	Fast	Slow	Fast	Slow	Fast	Slow
Eccentric						
Hip	2.24 ± 1.15	2.49 ± 1.02	5.77 ± 2.09	8.82 ± 4.7	7.95 ± 5.68	6.27 ± 3.10
Knee	15.52 ± 6.04	12.83 ± 4.59	13.63 ± 3.75	11.71 ± 2.64	14.04 ± 5.59	13.18 ± 4.55
Ankle	24.71 ± 6.13	20.18 ± 6.36	34.33 ± 16.78	24.09 ± 11.03	16.22 ± 6.57	26.06 ± 8.96
Concentric						
Hip	3.50 ± 1.65	2.40 ± 1.79	4.08 ± 1.33	3.12 ± 1.00	4.16 ± 1.79	4.02 ± 1.62
Knee	6.99 ± 1.65	6.30 ± 1.46	8.90 ± 3.98	7.63 ± 1.69	9.12 ± 3.39	8.43 ± 3.26
Ankle	7.13 ± 1.58	5.61 ± 1.68	7.57 ± 1.69	7.14 ± 1.63	8.92 ± 2.92	8.28 ± 3.64

## Data Availability

The original contributions presented in this study are included in the article. Further inquiries can be directed to the corresponding authors.

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
