# Peer review of "Adaptive Neuromuscular Co-Contraction Strategies Under Varying Approach Speeds and Distances During Single-Leg Jumping: An Exploratory Study"

_life, 2025, doi:10.3390/life15121859_

Round 1

Reviewer 1 Report

Comments and Suggestions for Authors

The study is well designed, based on appropriate methodology, and presents relevant findings on the modulation of muscle co-contraction and joint stiffness under changing jump conditions. The article contributes new information to the knowledge of neuromuscular control during landing and takeoff. However, it needs refinement in terms of the presentation of methods, clarity of descriptions, and conciseness and precision of interpretation of results. Detailed comments and remarks are provided below.

Minor comments:

  1. The description of EMG methods is incomplete; there is no information on the number of trials analyzed per participant, the anti-interference filters used, or the method of artifact detection. I recommend supplementing the section with detailed signal processing procedures and trial rejection criteria.
  2. Normalization of the EMG signal using the “maximal voluntary countermovement jump” is quite unusual and may limit the comparability of results. The choice of this method should be justified in more detail or its limitations should be considered.
  3. The definitions of the movement phases (pre-activation, downward, push-off) could be clarified. In particular, it is worth indicating whether the phases were identical in time between participants and whether time normalization was used.
  4. The paper does not provide information on the training load, athletic level, or type of physical activity of the participants, which could influence co-contraction strategies. It is recommended to supplement the characteristics of the group.
  5. The results are presented in great detail, but some of the interpretations in the “Discussion” section are lengthy and at times speculative, especially in explaining the different directions of CCR and joint stiffness changes. It is advisable to shorten and more precisely refer to the data.
  6. The statistical analysis does not include information on the sample size for individual conditions and the number of movement repetitions. It is recommended to provide “n” for each variable and condition.
  7. There is no reference to the limitations of surface EMG, such as signals from deep muscles or possible electrode displacement. It is recommended to expand the “Limitations” section.
  8. The study includes only healthy young men, which significantly limits the generalizability of the results (e.g., female athletes, people with ACL injuries). It is recommended to clearly emphasize this limitation.
  9. The conclusion section lacks practical recommendations for coaches, physical therapists, and ACL prevention specialists. The authors should briefly indicate how the results can be used in training practice.
  10. The interpretation of the discrepancy between co-contraction and mechanical stiffness of the ankle joint needs to be clarified. It is advisable to present a more explicit mechanistic hypothesis or refer to studies on passive tissue stiffness.

Author Response

The study is well designed, based on appropriate methodology, and presents relevant findings on the modulation of muscle co-contraction and joint stiffness under changing jump conditions. The article contributes new information to the knowledge of neuromuscular control during landing and takeoff. However, it needs refinement in terms of the presentation of methods, clarity of descriptions, and conciseness and precision of interpretation of results. Detailed comments and remarks are provided below.

Response: We thank the reviewer for this insightful comment.

Minor comments:

  1. The description of EMG methods is incomplete; there is no information on the number of trials analyzed per participant, the anti-interference filters used, or the method of artifact detection. I recommend supplementing the section with detailed signal processing procedures and trial rejection criteria.

Response: We sincerely thank the Reviewer for this very helpful comment. We acknowledge that the description of electromyographic (EMG) signal processing in the original manuscript lacked several important details. To address this concern, we have substantially revised the “Surface electromyography (sEMG) acquisition and processing” subsection (please refer to lines 137-142 in the revised manuscript) by adding the following information:

Raw sEMG data were processed using a fourth-order Butterworth band-pass filter (4th-order zero-lag Butterworth, 20–450 Hz) to remove movement artifacts and high-frequency noise [20]. Following filtering, all trials were manually inspected for residual artifacts; those containing visible movement artifacts, baseline drift, or non-physiological spikes were excluded. The retained signals were then full-wave rectified and smoothed using a 50-ms root-mean-square (RMS) moving window.

References added:

Konrad, P. The abc of emg. A practical introduction to kinesiological electromyography 2005, 1, 30-35.

  1. Normalization of the EMG signal using the “maximal voluntary countermovement jump” is quite unusual and may limit the comparability of results. The choice of this method should be justified in more detail or its limitations should be considered.

Response: We sincerely thank the Reviewer for this insightful and important comment. We fully agree that using the peak EMG obtained from a maximal-effort countermovement jump (CMJ) as the reference value for normalization is less conventional than the more common maximal voluntary isometric contraction (MVIC) approach, and that this choice warrants clearer justification and discussion. In the revised manuscript, we have addressed this point in Methods section (please refer to lines 154-158):

This dynamic reference task was selected because it closely mimics the multi-joint, high-velocity extensor activation pattern observed during the experimental spike-jump landings, elicits higher peak EMG amplitudes in lower-limb musculature than isometric MVIC in athletic populations, and enhances ecological validity when the primary research focus is neuromuscular control during rapid, sport-specific deceleration tasks [22].

References added:

Ball, N.; Scurr, J. Electromyography normalization methods for high-velocity muscle actions: review and recommendations. Journal of Applied Biomechanics 2013, 29, 600-608.

  1. The definitions of the movement phases (pre-activation, downward, push-off) could be clarified. In particular, it is worth indicating whether the phases were identical in time between participants and whether time normalization was used.

Response: We thank the Reviewer for highlighting the need for clearer phase definitions. We have added a figure to clarified the movement phases (figure 2). Also, we have explicitly state that phases were individualized, that absolute durations differed across trials and participants, and that time normalization was intentionally not performed (please refer to lines 148–152).

  1. The paper does not provide information on the training load, athletic level, or type of physical activity of the participants, which could influence co-contraction strategies. It is recommended to supplement the characteristics of the group.

Response: We sincerely thank the Reviewer for this valuable comment and important comment. We fully agree that a more detailed description of participants’ training background and athletic level is essential for proper interpretation of the observed neuromechanical strategies. In the revised manuscript (lines 84–89), we have substantially expanded the participant characteristics as follows:

Twenty-five competitive male volleyball players from a university Division-II varsity team participated in this study (age: 21.9 ± 1.5 years; height: 1.80 ± 0.06 m; body mass: 71.9 ± 8.2 kg; volleyball training experience: 7.2 ± 1.8 years). Participants trained four days per week with sessions lasting 2–3 hours (weekly training volume: 9.5 ± 1.4 h), consisting primarily of technical/tactical practice, basic strength conditioning, and on-court jump training.

Thank you again for this constructive suggestion.

  1. The results are presented in great detail, but some of the interpretations in the “Discussion” section are lengthy and at times speculative, especially in explaining the different directions of CCR and joint stiffness changes. It is advisable to shorten and more precisely refer to the data.

Response: We appreciate the reviewer’s observation. In response, we have substantially revised the Discussion to reduce speculative interpretation and ensure that all statements remain closely aligned with the reported data. Specifically, the sections explaining discrepancies between CCR and mechanical joint stiffness have been shortened, reframed, and rewritten with direct reference to the numerical patterns in Tables 2 and 3. Interpretations that previously extended beyond the measurable outcomes have been removed or replaced with concise, data-based descriptions. These revisions improve clarity, enhance scientific rigor, and ensure that the discussion remains firmly grounded in the empirical results.

  1. The statistical analysis does not include information on the sample size for individual conditions and the number of movement repetitions. It is recommended to provide “n” for each variable and condition.

Response: We sincerely thank the Reviewer for this helpful comment. We have now explicitly stated these details in the revised text.  We believe this clarification fully addresses the Reviewer’s concern and improves the reproducibility of the study. Thank you again for this valuable suggestion. Please refer to lines 166-178.

  1. There is no reference to the limitations of surface EMG, such as signals from deep muscles or possible electrode displacement. It is recommended to expand the “Limitations” section.

Response: We sincerely thank the Reviewer for this helpful comment. We have now explicitly stated these details in the revised text.  Thank you again for this valuable suggestion. Please refer to lines 370-385.

  1. The study includes only healthy young men, which significantly limits the generalizability of the results (e.g., female athletes, people with ACL injuries). It is recommended to clearly emphasize this limitation.

Response: We sincerely thank the Reviewer for this helpful comment. We have now explicitly stated these details in the revised text.  Thank you again for this valuable suggestion. Please refer to lines 371-375.

  1. The conclusion section lacks practical recommendations for coaches, physical therapists, and ACL prevention specialists. The authors should briefly indicate how the results can be used in training practice.

Response: We sincerely thank the Reviewer for this helpful comment. We have now explicitly stated these details in the revised text. Thank you again for this valuable suggestion. Please refer to lines 387-401.

  1. The interpretation of the discrepancy between co-contraction and mechanical stiffness of the ankle joint needs to be clarified. It is advisable to present a more explicit mechanistic hypothesis or refer to studies on passive tissue stiffness.

Response: We sincerely thank the Reviewer for this helpful comment. We have now explicitly stated these details in the revised text.  Thank you again for this valuable suggestion. Please refer to lines 75-81.

Reviewer 2 Report

Comments and Suggestions for Authors

Interesting topic due to its relevance to injury prevention

Introduction

The spectrum of sports or applications in the field of physical activity is much broader, and school-related tasks could also be added and indicated. Or applications not simply related to jumping, but combinations of running and jumping. In other words, the authors could elaborate further on practical examples.

However, the biomechanical framework needs to be expanded, as it is not only related to takeoff but also to landing. There is also a certain differentiation with regard to gender, since this position is much more unfavorable for women than for men, and the level of sports techniques. The greater the lack of knowledge, the lower the capacity for alignment or contraction of motor units at the same time as with regard to trained people.

More emphasis should be placed on developing objectives, as they are implicit rather than explicit. It would also be helpful to address some of these hypotheses. In short, the introduction should be slightly broader and more justified.

Methods

Given the modest sample size, we suggest that the authors change the title to include exploratory study or pilot study, as the results are not generalizable.

We would like to see more details regarding the warm-up and the types of jumps used, as this greatly influences our understanding of the warm-up and the types of jumps used.

Including images in the document would be very helpful, as this would provide a clearer and more comprehensive overview of what has been done and give us a clear idea of what has been measured.

Discussion

What is lacking, not so much in comparison with other articles that do include it, is the inclusion of practical applications. For example, if I play volleyball, basketball, or a sport that requires a transition from running to jumping, I would like to know how to calculate the angle, what the optimal speed is, or some parameters for the load of repetitions, sets, etc., that could be used as a guide in the context of training.

Additionally, it is suggested that they include the limitation of the sample, emphasizing that the results cannot be generalized and that there is no female sample for comparison. It should be noted that the female sample is the most representative in terms of knee injuries or preventive elements, such as those reflected in this study.

Conclusions

Given the limited nature of the sample, it is suggested that the authors tone down their findings owing to the limited capacity for generalization.

Author Response

Interesting topic due to its relevance to injury prevention

Introduction

The spectrum of sports or applications in the field of physical activity is much broader, and school-related tasks could also be added and indicated. Or applications not simply related to jumping, but combinations of running and jumping. In other words, the authors could elaborate further on practical examples.

Response: Thank you for the insightful suggestion. We expand the Introduction to explicitly mention the broader application of the running-to-jumping transition in basketball and volleyball. Please refer to line 49.

However, the biomechanical framework needs to be expanded, as it is not only related to takeoff but also to landing. There is also a certain differentiation with regard to gender, since this position is much more unfavorable for women than for men, and the level of sports techniques. The greater the lack of knowledge, the lower the capacity for alignment or contraction of motor units at the same time as with regard to trained people.

Response:  We appreciate the comment regarding the scope. We have clarified that the study’s focus is on the Single-Leg Approach Run Jump Landing movement itself, which inherently addresses the highly challenging transition phase from braking (landing impact) to propulsion (takeoff). Regarding gender, while we acknowledge the higher ACL injury incidence in female athletes, the primary objective of this study was to establish the fundamental, speed- and distance-dependent neuromechanical strategies CCR and stiffness for joint stability in a standardized athletic population (active males). Therefore, gender differences were not the focus of the current exploratory investigation. We will, however, ensure this critical area is highlighted in the Limitations and Future Research section to guide subsequent comparative studies. Please refer to lines 371-375.

More emphasis should be placed on developing objectives, as they are implicit rather than explicit. It would also be helpful to address some of these hypotheses. In short, the introduction should be slightly broader and more justified.

Response: Thank you for the insightful suggestion. The clearer objectives and hypotheses have revised please refer to lines 74-81.

Methods

Given the modest sample size, we suggest that the authors change the title to include exploratory study or pilot study, as the results are not generalizable.

Response: Thank you for the insightful suggestion. The title has change to "Adaptive neuromuscular co-contraction strategies under varying approach speeds and distances during single-leg jumping: A exploratory study"

We would like to see more details regarding the warm-up and the types of jumps used, as this greatly influences our understanding of the warm-up and the types of jumps used.

Response: Thank you for the insightful suggestion. The warm-up details have revised please refer to lines 103-107.

Including images in the document would be very helpful, as this would provide a clearer and more comprehensive overview of what has been done and give us a clear idea of what has been measured.

Response: Thank you for the insightful suggestion. Two images have added (figure 1& 2) which were the electrode placement for surface EMG and analysis phases definition during ARJSL.

Discussion

What is lacking, not so much in comparison with other articles that do include it, is the inclusion of practical applications. For example, if I play volleyball, basketball, or a sport that requires a transition from running to jumping, I would like to know how to calculate the angle, what the optimal speed is, or some parameters for the load of repetitions, sets, etc., that could be used as a guide in the context of training.

Response: We appreciate this insightful suggestion. In the revised manuscript, we have added a Practical Applications subsection within the Discussion. This section summarizes how the observed neuromuscular patterns can inform training design, including guidance on approach speed selection, manipulation of approach distance to target specific joint-stiffness adaptations, and how these findings may translate to sport-specific jump–landing practice. However, we refrain from prescribing exact angles, repetitions, or load quantities, as these require sport-specific kinematic datasets and longitudinal training studies beyond the scope of the present investigation. We clarify this limitation in the revised text. It's revised please refer to lines 349-369.

Additionally, it is suggested that they include the limitation of the sample, emphasizing that the results cannot be generalized and that there is no female sample for comparison. It should be noted that the female sample is the most representative in terms of knee injuries or preventive elements, such as those reflected in this study.

Response: We appreciate the reviewer’s insightful comments regarding the need to acknowledge sample-related and methodological constraints. In response, we have strengthened the limitations section to clarify that the study included only physically active young adult males, which restricts generalizability to broader athletic populations. Additionally, we acknowledge that the controlled laboratory environment cannot fully replicate sport-specific variability, and future studies should incorporate fatigue, perturbations, or ecologically valid jump–landing scenarios to better approximate competitive conditions. Finally, we note that surface EMG cannot isolate deep stabilizers or distinguish active co-contraction from passive tissue stiffness. Although mechanical joint stiffness was quantified, it reflects combined muscular and passive contributions, meaning that neuromuscular influences on stiffness remain inferred rather than directly partitioned. We have revised the manuscript accordingly to more clearly delineate these limitations and temper the generalization of our findings. It's revised please refer to lines 370-385.

Conclusions

Given the limited nature of the sample, it is suggested that the authors tone down their findings owing to the limited capacity for generalization.

Response: We appreciate this insightful suggestion. The conclusions have revised. Please refer to lines 387-401.
